# High-Precision Single-Photon Laser Time Transfer with Temperature Drift Post-Compensation

**DOI:** 10.3390/s20226655

**Published:** 2020-11-20

**Authors:** Wendong Meng, Yurong Wang, Kai Tang, Zhijie Zhang, Shuanggen Jin, Ivan Procházka, Zhongping Zhang, Guang Wu

**Affiliations:** 1State Key Laboratory of Precision Spectroscopy, East China Normal University, Shanghai 200062, China; wdmeng@shao.ac.cn (W.M.); yrwang152@163.com (Y.W.); 2Shanghai Astronomical Observatory, Chinese Academy of Sciences, Shanghai 200030, China; kaitang@shao.ac.cn (K.T.); zjzhang@shao.ac.cn (Z.Z.); sgjin@shao.ac.cn (S.J.); zzp@shao.ac.cn (Z.Z.); 3Department of Physical Electronics, Faculty of Nuclear Sciences and Physical Engineering, Czech Technical University in Prague, 115 19 Prague, Czech Republic; ivan.prochazka@fjfi.cvut.cz

**Keywords:** laser time transfer, single-photon detection, temperature drift compensation, avalanche photodiodes, laser ranging

## Abstract

Laser time transfer is of great significance in timing and global time synchronization. However, the temperature drift may occur and affect the delay of the electronics system, optic generation and detection system. This paper proposes a post-processing method for the compensation of temperature-induced system delay, which does not require any changes to the hardware setup. The temperature drift and time stability of the whole system are compared with and without compensation. The results show that the propagation delay drift as high as 240 ps caused by temperature changes is compensated. The temperature drift coefficient was diminished down to ~0.05 ps/°C from ~20.0 ps/°C. The system precision was promoted to ~2 ps from ~11 ps over a time period of 80,000 s. This method performs significant compensation of single-photon laser time transfer system propagation drift and will help to establish an ultra-stable laser time transfer link in space applications.

## 1. Introduction

The ground-space laser time transfer with high resolution and high accuracy is of great significance in improving timing and global time synchronization [1,2,3]. The basic principle of laser time transfer relies on the light pulse propagation time measurement between clocks at different locations to be synchronized. There are two typical two-way laser time transfer schemes. One is effectively a two-way time transfer, whereby the laser is emitted from both ground and space segments to each other. In this system, both locations will emit and receive the laser pulses, record their time tags and exchange the above information to calculate the time difference and distance between clocks. The other is also two-way, but one location (usually the ground station) will emit laser pulses, and the other (usually the space segment) will only receive them. In addition, the laser reflector array (LRA) on board reflects a part of the laser pulse back to the laser-emitting station for exact two-way propagation time measurement. The LRA is a totally passive optical component whose contribution to laser time transfer performance is not affected by the space environment. This allows the onboard instruments to be much more compact than the former. All of the existing and planned ground–space laser time transfer projects are now based on this kind of structure [2,3,4,5,6], such as the Chinese Laser Time Transfer (LTT), European Time Transfer by Laser Link (T2L2) and the ongoing European Laser Timing (ELT), Chinese Laser Timing (CLT) system onboard Chinese Space Station (CSS) and European Laser Timing + (ELT+) in space optical clock on International Space Station (I-SOC).

The existing laser time transfer link T2L2 together with the onboard clocks has achieved a limiting time transfer precision better than 10 ps for integration time of 10 to 100 s, and 10 ps for integration time of several days [7]. The ongoing ELT project onboard the International Space Station (ISS) should provide an ultimate precision of 3 ps for 300 s of integration time with a clock stability of 10^−16^ [8]. With the development of optical clocks, a frequency stability of 10^−17^ was provided in several ground-based labs. The launch of space optical clocks is now under preparation [5,9]. Optical clocks may approach a frequency stability of 10^−18^ when operated in space. The time and frequency transfer of high stability between ground and space presents a fundamental experiment in physics. Laser time transfer ultimate precision, expressed by time deviation (TDEV) [10], is expected to be better than 0.3 ps over days in order to compare the frequency from the ground to space with 10^−18^ precision [11]. The next goal of the laser time transfer system is now planned in the I-SOC mission by Europe and CLT mission by China to achieve a time stability of approximately 0.5 ps over 100 s and 1 ps over 10^6^ s with optical clocks [5,9].

With such high-precision requirements, both ground and space laser time transfer systems have great challenges. The onboard optics and instruments in space need to be especially well designed to achieve high accuracy and stability. In low earth orbit, the temperature on a satellite surface may change more than 100 °C. With some insulation methods, tens of degrees Celsius temperature variations may occur at the space-receiver side. The temperature changes have a strong effect on the delay of the laser time transfer chain. Therefore, it is important to control and compensate the system temperature drift to achieve better time stability performance [12].

The existing satellite laser ranging (SLR) technology was optimized to control the system temperature coefficient of a single instrument. The single-photon detector package based on a 25-μm K14 chip from Czech Technical University achieved a temperature coefficient of 0.76 ± 0.1 ps/°C [13]. The event timer to record the time tags of the laser pulses has a drift of less than 2 ps/°C [14]. The coaxial cable drift is within the range of 0.05 to 5 ps/°C per meter [11,15]. The laser pulses emitting time also changes several picoseconds per Celsius degree. SLR stations keep all the instruments in a temperature-stable room and use self-calibration to maintain the stability of the measurements. However, the onboard laser time transfer payload temperature cannot be controlled so well. This is why the temperature coefficient of the whole system should be studied and described. This will enable scientists to compensate most of the delay drifts and achieve better time stability.

In this paper, a basic laser timing test setup is established and the system delay as a function of temperature is measured. System temperature dependence is analyzed as a whole and a temperature drift post-compensation method is proposed to compensate for the system delay drift. Applying this method, the onboard system delay can be tested and described before the launch. In space operation, it can be modeled and calibrated according to the operating temperature. Section 2 and Section 3 present the method and experiments, results and analysis are shown in Section 4, and finally, conclusions are given in Section 5.

## 2. Temperature Drift Experiment Setup

The laser time transfer system is a process of laser-pulse detection and epoch recording both on board a satellite and at an on-the-ground station. Due to the effects listed above, we focused on the space segment of the system. The experiment system of space segment contains a laser, a single-photon avalanche detector (SPAD) and a timing system (see Figure 1). A short-pulsed semiconductor laser at 709 nm with a pulse width of 70 ps was used as the light source. It was triggered by a two-channel signal generator and synchronized with the gate signal to the SPAD. The gate signal is used to set the SPAD into its sensitive mode. The first detected signal of the SPAD will reset it as unsensitive again. The repetition rate of the entire experiment was set to 10 kHz. The laser beam was collimated by a telescope system composed of two lenses marked as L1 and L2. An aperture was placed at the focus of the two lenses for spatial filtering. After attenuation, the laser beam was coupled into a multi-mode fiber (2 m long) by a lens with a focal length of 50 mm. The fiber output was coupled to the SPAD active area. The number of incident photons could be adjusted consecutively by adjusting the continuous attenuator or changing the distance between the SPAD and the output head of the multi-mode fiber.

In this experiment, the SPAD detector was operated in an active quenching and gating mode. A detection chip with a 25-µm-diameter active area (K14 series) was used [16]. It was produced by Czech Technical University and has been used in the onboard LTT instrument on the Chinese Beidou Navigation system [2]. A single-shot precision of 150 ps was achieved with this application. Thanks to the high-speed comparator and faster Resistor-Capacitance (RC) circuit, with the same chip, our SPAD detector achieved a single-shot precision better than 30 ps. We set a bias voltage of 2 V above the SPAD’s breakdown voltage to get a detection efficiency of approximately 30% at 709 nm. An event timer (ET) with a precision of 8 ps and resolution of 13 ps was used to record the propagation delay changes of the entire transfer chain. A multi-channel temperature recorder (TR) (NAPUI TR230X) based on thermocouple was used to record the temperature changes over time with a precision of 0.1 °C. A thermocouple was set on the bottom of the SPAD near the optical platform representing the room temperature. At first, two data sets were recorded simultaneously: propagation delay as a function of time (by ET) and temperature as a function of time (by TR). Then, all the collected data were processed. A curve of propagation delay as a function of temperature was obtained, the fitting equation was calculated and then the temperature compensation function was established. Finally, the variation of original propagation delay results was obtained using the compensation function.

The laboratory temperature was changed to simulate the temperature variations in the space environment. The initial room temperature of 22 °C was slowly increased by 2–3 degrees per day, and data were collected for 7 days till 34 °C. In the data processing, we used a “2.2 × σ” editing procedure to extract the useful signal [17]. Furthermore, we analyzed the whole system temperature drift and time stability in terms of TDEV. Usually, original data record length should be at least 3 times of the average time, which are uesed to analyze time stability [10]. Considering the actual measurement time and the upload transfer speed of ET in this test, we chose 10 kHz as the laser frequency and a 10% counting rate, which corresponds to an incident light intensity of 0.3 photon/pulse. This setup simulated the in-space operation.

## 3. Experimental Data and Processing Methods

With the above experiment setup, we finally achieved a single-shot precision of 45 ps expressed in terms of the root mean square (RMS) of the entire chain. The raw data are illustrated in Figure 2a, revealing in general the same trend of variation between the propagation delay and the temperature. According to the data collected at different temperatures, one could get the delay time by recognizing the peak position. Figure 2b shows a delay change of 240 ps as the temperature changes from 22 °C to 34 °C. According to Figure 2, the whole temperature changing process can be divided into 7 sections. First comes a temperature stable period, which is followed by 6 temperature rising and stable periods. Each temperature rising period lasts 7–10 h, while the temperature stable period lasts 12 to 15 h. However, at the end of day 3, some errors occurred in the data recording system, so only 2 h of data for the temperature stable period were recorded that day.

Figure 3 illustrates the trend of propagation delay along with the temperature variation. According to Figure 3, we assume the relationship of system propagation delay to temperature as a one-order linear change.

The raw data collected at different temperatures are represented by $d_{i}$. One could get the delay time at different temperatures by recognizing the peak position or the mean value, which can be expressed by *D**_j_*. The delay as a function of temperature can be plotted and linearly fitted as
(1)d=aT+b
where *d* is the ideal delay time without compensation in unit of ps, *T* is the temperature in unit of °C and *a* is the slope of the fitting line curve. In principle, the variation of original propagation delay could be compensated to a value of temperature *T*_0_*,* which could be selected freely. The compensated delay time *d_c_**_i_* can be calculated by
(2)$$d_{ci}=d_{i}-\left(T_{i}-T_{0}\right)\times a$$
where $T_{i}$ is the corresponding temperature of data $d_{i}$. With the above experiment system and compensation method, we recorded all the data, took part of the data to get a fitting curve and the function and finally got the compensated data. Using the original and compensated data, we calculated and analyzed the time stability of the system before and after compensation.

Though the system delay varies linearly with the temperature, we see some fluctuations when the temperatures begin to change. The most likely reason is that the temperature will take time to reach a steady state. This is why we separated the data into two groups corresponding to temperature stable and rising conditions for data analysis.

## 4. Results and Discussion

### 4.1. Data Processing and Analysis

#### 4.1.1. The Temperature Stable Period

Each section of the temperature stable periods lasted for 12–15 h except for the third day with only approximately 2 h. We extracted data for each day with about half an hour to calculate the typical propagation delay in specific temperatures, and got a fitting equation as
(3)d=20.8T+32651.2

If we want to compensate all the data to 25 °C, the compensated ideal propagation delay $d_{c0}$ can be calculated as 33170.7 ps.

The compensated delay *d_c_**_i_* can be calculated by
(4)$$d_{ci}=d_{i}-20.8T+519.5$$

All of the compensated data are calculated and shown in Figure 4, and the slope of the delay changed to almost flat as −0.034 ps/°C. With the raw and compensated data, we got the TDEV before and after compensation. Figure 5 shows the stability described above, where the black dots and lines represent the raw data results, and the red dots and lines represent the compensated data results. For approximately a half-day averaging time, TDEV achieves 1.16 ps from the original value of 16.2 ps (over 40,000 s). The improvement by temperature effect compensation is more than one order of magnitude.

#### 4.1.2. The Temperature Rising Period

The data processing of the temperature rising period data is the same. The liner fitting curve can be expressed by
(5)d=20.4T+32672.3

At 25 °C, the ideal propagation delay will be 33182.4 ps.

The compensated delay *d_c_**_i_* can be expressed by
(6)$$d_{ci}=d_{i}-20.4T+510.1$$

All of the compensated data are shown in Figure 6.

The slope of the delay after compensation achieves −0.054 ps/°C. The comparison of TDEV is shown in Figure 7, which also shows better stability after compensation. The long-term stability achieves 1.76 ps from the original value of 7.74 ps for averaging time of 40,000 s.

#### 4.1.3. Analysis of the Combined Data

The above compensated data are combined to an entire experiment process as shown in Figure 8a. The propagation delay drift of 240 ps has been removed almost perfectly.

We also got the stabilities of the combined data with and without compensation as shown in Figure 8b. One can see from the TDEV diagram that the short-term stability gets worse in the averaging time between approximately 20 s and 1300 s; however, the long-term stability is much lower than approximately half an order of magnitude from the original value of 11.5 ps down to 2.5 ps for averaging time of 80,000 s.

### 4.2. Discussion

From the TDEV analyzed above, one can conclude that the linear-fitting compensation method works better with the data in the temperature stable period. In the temperature rising period, the propagation delay is a little complex (see Figure 6). This demonstrates the complexity of thermal conduction in the test system and the instruments. Although the compensation method is not perfect, the 240 ps delay change was nearly perfectly compensated and the detection delay coefficient was reduced from the original value of 20 ps/°C to −0.05 ps/°C. The time stability expressed in a form of time deviation (TDEV) was also greatly improved. In the temperature stable period shown in Figure 5, the short-term time stability after the averaging time of 100 s flattens down and achieves 1.2 ps from the original value of 16 ps over 40,000 s. In the temperature rising period, due to the different heating and thermal transition conditions at different parts of the system, the temperature changing is not uniform, and the one-order linear compensation is not sufficient. Nevertheless, the long-term time stability still improved by half an order of magnitude, from the original value of 6.8 ps to 2.1 ps over 40,000 s. Although the TDEV of compensated data is not perfect, this post-compensation method is effective for the delay drift of the whole laser time transfer system.

The experiment is now carried out in the atmosphere where the heat is mainly transmitted through the air. The instruments are separated and have low heat transport within each other. In the space application, the entire system will be very small and compact. This is why their temperatures will be much closer to each other. In addition, the entire instrument will be thermally insulated by thermal cladding, which will help to reduce temperature changes and hence the system delay drift. Considering these facts, the temperature drift post-compensation will be more efficient.

## 5. Conclusions

In this paper, we established a laser time transfer experiment setup to test the system propagation delay in a room temperature tuning range from 22 °C to 34 °C and proposed a temperature drift post-compensation method to compensate the system propagation delay. As a result, the original system propagation delay drift of 240 ps caused by temperature change was reduced by compensation of more than two orders of magnitude. The temperature drift coefficient of detection delay was decreased from the original value of 20 ps/°C down to −0.05 ps/°C. The time deviation of detection delay in the temperature stable period was improved from the original value of 16 ps down to 1.2 ps over 40,000 s. In the complete experiment, the time deviation was reduced down to 2.5 ps from the original value of 11.5 ps over 80,000 s. Thus, the post-compensation method used in this experiment performed a significant compensation for the system delay drift. The corresponding TDEV long-term stability was increased by a 0.5–1 order of magnitude. With a compact design and other methods such as thermal isolation, this post-compensation method will help to establish an ultra-stable laser time transfer system for space application.

## Figures and Tables

**Figure 1 sensors-20-06655-f001:**
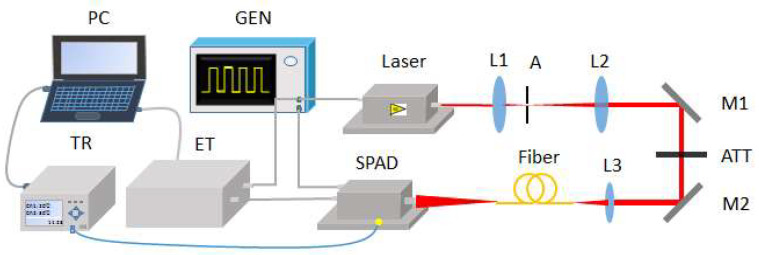
Experimental setup of the laser time transfer based on the temperature drift post-compensation. GEN is a generator; L1, L2 and L3 are planoconvex lenses; A represents the aperture; M1 and M2 are high-reflection mirrors; ATT is an attenuator; SPAD is a single-photon detector based on Si; ET is an event timer (precision: 8 ps); and TR is a temperature recorder.

**Figure 2 sensors-20-06655-f002:**
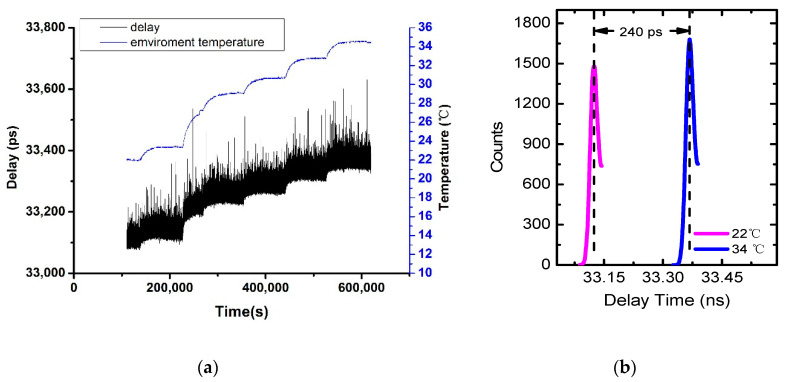
(**a**) The propagation delay and room temperature over the time. (**b**) The propagation delay drift of environment temperatures from 22 °C to 34 °C.

**Figure 3 sensors-20-06655-f003:**
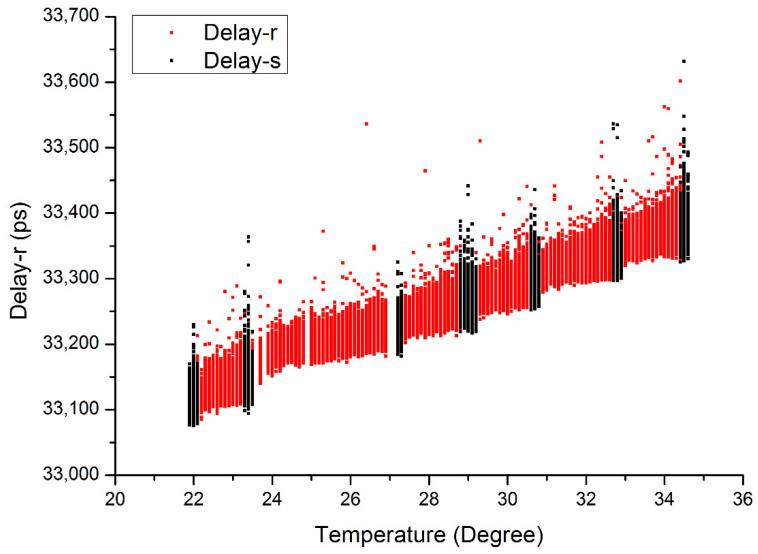
Propagation delay vs. environment temperatures. (The black dots represent the data of the temperature stable period and the red dots represent the temperature rising period).

**Figure 4 sensors-20-06655-f004:**
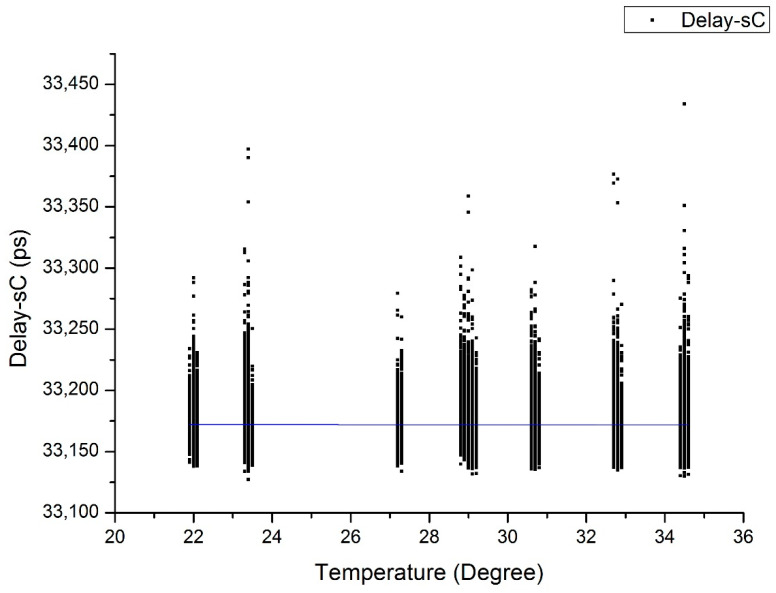
Delay over temperatures after compensation in the temperature stable period. (The black dots are delay after compensation, and the blue line is the one-order fitting curve).

**Figure 5 sensors-20-06655-f005:**
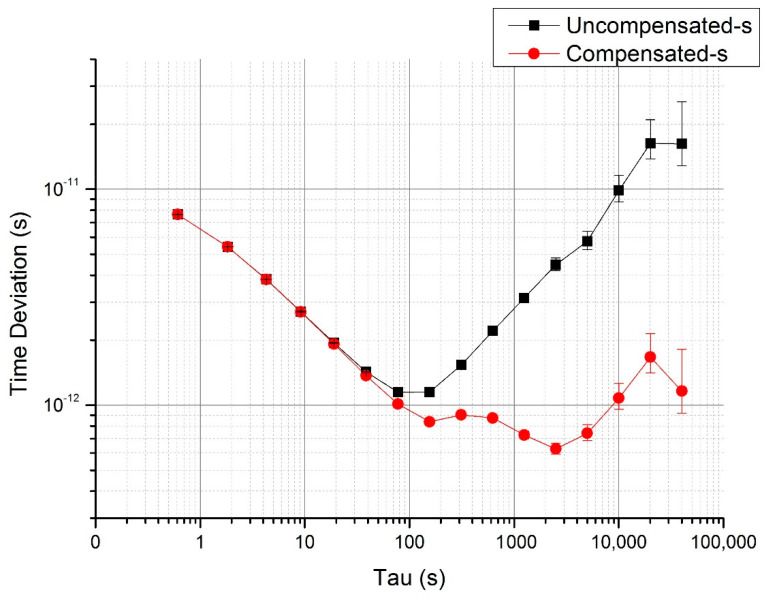
Time deviation (TDEV) before and after compensation in the temperature stable period.

**Figure 6 sensors-20-06655-f006:**
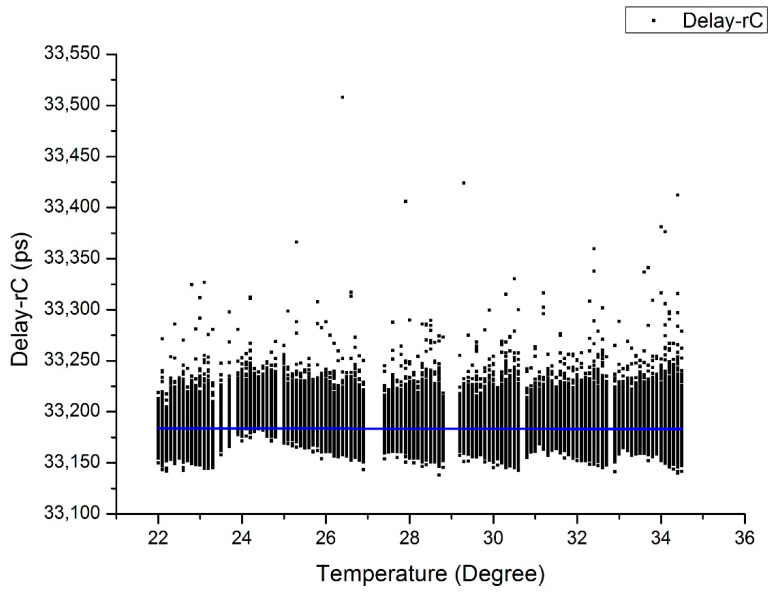
Delay over temperatures after compensation in the temperature rising period. (The black dots are the compensated data, and the blue line is the fitting curve).

**Figure 7 sensors-20-06655-f007:**
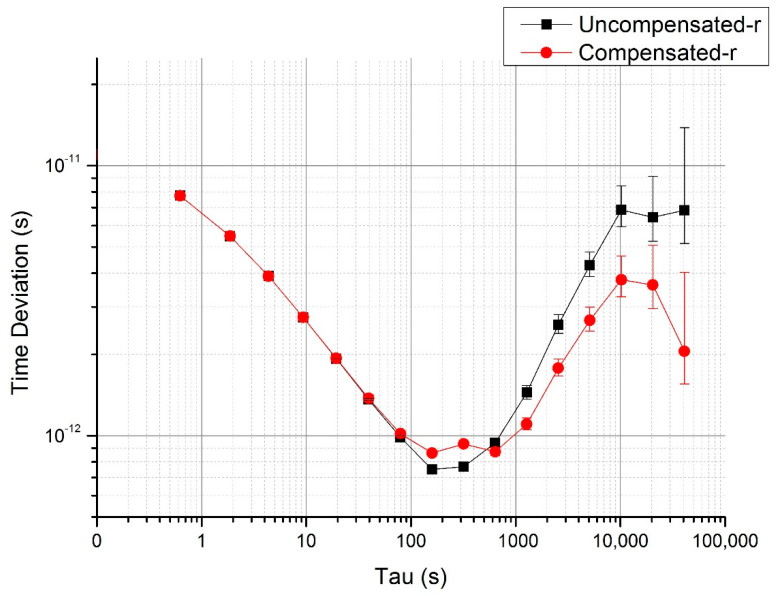
TDEV before and after compensation in the temperature rising period.

**Figure 8 sensors-20-06655-f008:**
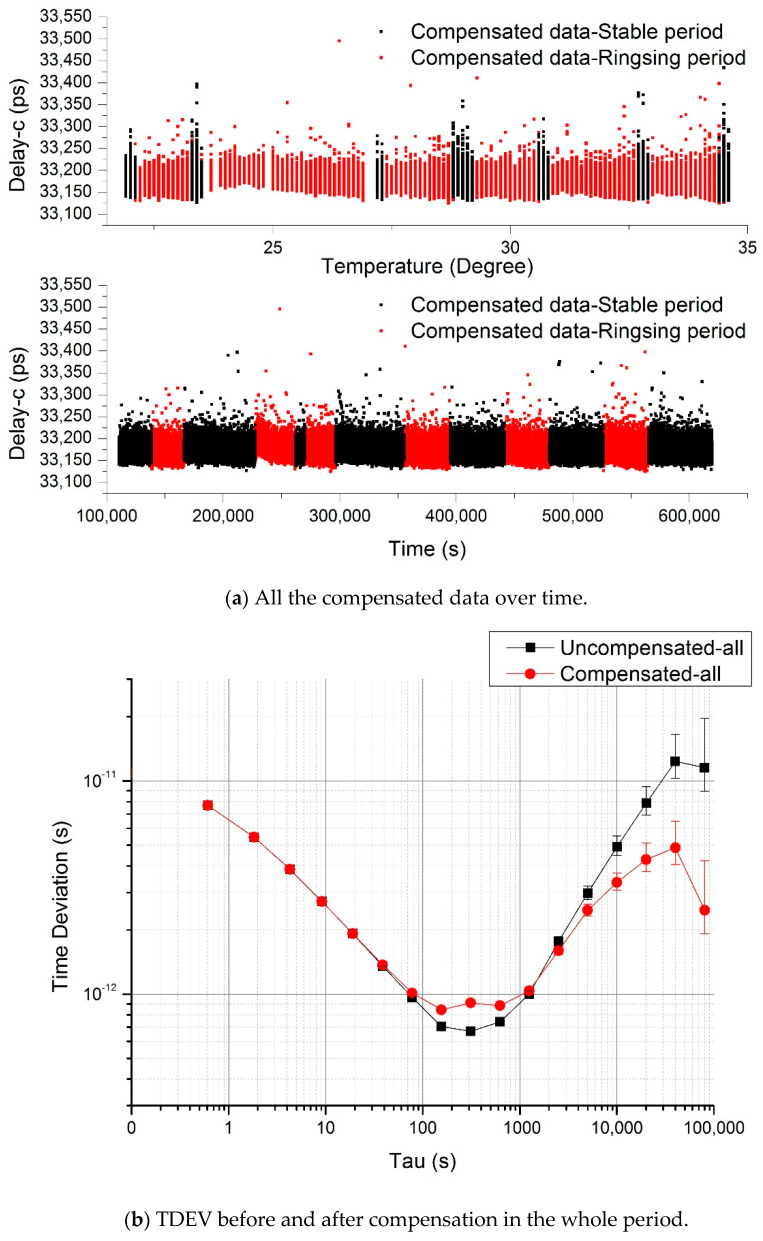
(**a**) All the compensated data over time. The upper panel is delay vs. temperature, and the lower panel is delay vs. time. (**b**) TDEV before and after compensation in the whole period.

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
