# Peer review of "High-Precision Single-Photon Laser Time Transfer with Temperature Drift Post-Compensation"

_sensors, 2020, doi:10.3390/s20226655_

Round 1

Reviewer 1 Report

Manuscript ID: sensors-940957
Type of manuscript: Letter
Title: High-precision Single-photon Laser Time Transfer with Temperature Drift Post-compensation
Authors: Wendong Meng, Yurong Wang, Kai Tang, Zhijie Zhang, Shuanggen Jin, 
Ivan Procházka, Zhongping Zhang, Guang Wu

This is an interesting paper on an important aspect of time-coherent spectroscopy, such as laser ranging and optical time transfer. Variable system delays, caused by temperature variation are a key issue in the time of flight measurement technique. The paper could be considered for publication when a few problems have been removed. These issues are listed below.

Language: Although the paper is well structured, the presentation is not yet adequate for the journal. There are numerous issues in forming sentences, the use of proper words/phrases and a lot of missing articles. One example of many is the sentence starting at line 16 in the abstract. It could probably be reorganized in the following way:

“This paper proposes a post processing method for the compensation of the temperature induced system delay, which does not require any changes to the hardware setup.”

In line 71 after the number 25 the units should be a Greek mu instead of the u. What are “better results” in line 80? I think the authors want better stability, right? What means “acknowledgement to the APD chip” in line 110?

I recommend to ask a native speaker to help in phrasing out the document.

Presentation: I think the paper would benefit from compacting it. There are a few redundant plots, which I suggest to remove. These are fig. 3 (the effect is already obvious from fig. 2(a), the lower part of fig. 4(b) as well as the lower part in fig. 6(b). The upper part of fig. 8(a) is also redundant. There is an issue with figure labeling in line 192. While fig. 5 is referenced, the sentence actually describes figure 6.

The authors use the symbol “t” in order to refer to temperature. I strongly recommend not to do that, as “t” is usually referring to time, which by the way also plays a role in the presented experiment. So that is really confusing. The proper way would be to use “T” for temperature in general and for the fixed reference temperature (used in line 166) an index like “0” or “I” would be appropriate.

Some of the description is repetitive, so I would recommend to sharpen the presentation of the work by shortening it at the same time. One example of several is the beginning of chapter 2, where the individual items of the experiment are listed (line 91). Subsequently they are described by their function (line 97/98).

Content: Although the presented results are quite promising, there are some shortcomings in the actual scientific content of the paper. In line 94/95 a modulation of the light source with the help of a two-channel signal generator is mentioned. What kind of modulation is this and what does it achieve? The paper clearly lacks some information here.

From line 120 onwards, there is a description of temperature measurements, but it is not stated in the document, neither indicated in fig. 1 where the temperature values have been taken.

For the reported experiment it appears to be the case that the entire setup as shown in fig. 1 has been subjected to the temperature variation between 22 and 34 degrees C. This would mean that there are many temperature related contributors to the overall detected signal delay. Some effect could come from the laser, some from the fiber (whose length is actually not stated). The detector could contribute and more seriously the event timer itself. So the simple linear correction model describes a correlation between temperature and delay, which is composed out of several sources, which may respond at slightly different timescales, for example due to different thermal mass of the individual delay contributors. This clearly explains, why the agreement is good for the time periods where the experimental setup is in thermal equilibrium and less good for the periods where the temperature transits from one stable state to another.

That leaves the reader wondering what is actually the take home message of this paper. The true source of the temperature related delay drift has not been identified, only the bulk effect was captured. Neither has the setup been packaged in a way that it would correspond to a space instrument. If several thermal sensors on all relevant hardware components had been used, it would probably allow to provide a good delay compensation during the periods of temperature variation, since different temperature coefficients and drift rates of the different contributors could be handled more specifically.

Reviewer 2 Report

1 Research design

One would like to further understand the motivation  (=value of) this paper why  this kind of post compensation test has significant help for time transfer accuracy was not comprehensively described in the paragraph (line #70~84 ).

1) What does it mean line#77~78 "but as to the onboard laser time transfer payload and system, the temperature coefficient has not specially controlled due to the previous application." ?

2) What do you mean "a simplified experiment" (line#251) you established ? In another words, what do you have complex experiment situation in your mind ?

2  Test method and results 

1) Experimental set up shown in Figure 1.

The position of temperature sensors is exactly same as one the mission you have by those telemetry data you read ?  If not, some assumption is needed reasonably understood to readers such as "close enough" or/and thermal model point of view in spacecraft structure.

2) In section 4, it is good point you had eye on separate modeling to phenomena of temperature between stable and rising period.  One believe  this test was done in atmosphere where convection affects  the results. One would like to have your comment in 4.4 Discussion section on the difference of result to the case of spacecraft where thermal conduction only exists.   

3 English/Typo things:

1) Line#160 and expression(1):Upper or lower case of  Di, DI, miss-usage ?

2) Line#165 and expression(2), Upper or lower case of  Dc, DC, miss-usage ?

3) In the expression(4), what is d in right hand term stands for, raw data ? Please give definition.

4) In Line#187, DIC means ideal calibrated delay example ?   Since DIC never appear in other parts. it is better to express  dC(at 25 degC) and please unify upper or lower case when you use them as the same thing.

5) Number of right hand ordinary axis is missing in Figure 4 (a) and  Figure 6(a).

6) In Line#184, Figure 4 must be Figure 4(b).

7) In Line#192, Figure 5 shows the above stability graphs--> Figure 5 shows stability described above.

8) In Line#194, the blue dots and green lines--> the blue dots and red lines (as one can see legend in Figure 5).

9) All the graph use colors such as red, blue, green and purple. Is this paper will be published as color recognized ?  If not, appropriate measure should be taken for the graph drawings.

Round 2

Reviewer 1 Report

The paper has considerably improved over the previous version, however a number of issues remain. Although indicated in the previous review, the use of the English language has not improved. There is still a very considerable number of articles missing all throughout the document. Furthermore, there are a number sentences, which need restructuring. One example is in line 144/145 and could possibly be rewritten as: “The raw data was collected and is shown in Figure 2(a), revealing in general the same trend of variation between the propagation delay and the temperature”.

I still have an issue with figure 9 and the conclusions from it: We agree on the fact that under stationary temperature conditions there is a linear relationship between temperature and the obtained delay. In the period where the setup transits into the next state there is not such a good agreement. I also accept that it may not be possible to capture the full effect by the addition of further temperature sensors on other components in the flight module, such like the timer, the signal conditioning (if present) or the laser.

However, for a paper like this, I think it is necessary to investigate the causes of the temperature variation properly. I do not think that there is only one single cause and that this is associated with the detector alone. The result of a more detailed assessment would be a much better correction model. Furthermore, it would identify the proper packaging of the time transfer experiment on board, which would be important for the space experiment too.  
